# The Impact of Research Results Feedback on the Lived Experiences of Elderly Participants in the DIMAMO Health Demographic Site: A Case of AWI-Gen Participants

**DOI:** 10.3390/ijerph22101565

**Published:** 2025-10-15

**Authors:** Reneilwe G. Mashaba, Cairo B. Ntimana, Precious Makoti, Katlego Mothapo, Joseph Tlouyamma, Kagiso P. Seakamela

**Affiliations:** 1DIMAMO Population Health Research Centre, University of Limpopo, Sovenga St, Polokwane 0727, South Africa; given.mashaba@ul.ac.za (R.G.M.); precious.makoti@ul.ac.za (P.M.); katlego.mothapo@ul.ac.za (K.M.); joseph.tlouyamma@ul.ac.za (J.T.); peacekagiso4@gmail.com (K.P.S.); 2Department of Pathology and Medical Sciences, University of Limpopo, Sovenga St, Polokwane 0727, South Africa

**Keywords:** results feedback, lived experiences, research participation

## Abstract

The concept of engaging the community in the results of research or returning individual results to the respective participants seems more of an afterthought than an integral part of research processes. The study aims to assess the impact of research results feedback among study participants from a rural black community of low socio-economic status. The study was qualitative. The interview data was analyzed using a deductive content analysis method to develop themes. The present study included about 31 individuals, of which 79% were women and 29% were men. The study used deductive content analysis to identify themes. These themes included health and lifestyle awareness, perceived benefits of research participation, community perception of health research, economic and social motivations, and challenges and concerns. The present study shed light on the importance of returning individual results for participants in health research. Providing feedback was found to improve participants’ health awareness, lifestyle behavior and contribute to early disease detection, especially for conditions that are not routinely tested in clinics. The themes that were generated showed that participants altered their lives and health-seeking behavior because of the information they received from participating in research, a situation that would not have happened if they did not have the results back. The findings of the present study indicate that it is important to return feedback results post conducting a research study. Returning results not only improves the livelihood of participants at the community level but also has the potential to foster strong researcher–community partnerships to enhance research participation and health outcomes, especially in disadvantaged populations.

## 1. Introduction

Health research has an impact on the lives of the individual participants and the community in which the research is conducted. The information generated through health research has an impact on how the communities organize themselves to improve their health outcomes [1,2]. Research structures were previously designed to make generalized arguments about the population studied, which has led to researchers not returning to study participants’ findings [3,4,5]. Topically, there is growing interest in the need to make research findings publicly available to individuals and stakeholders [6].

The ethical principles of autonomy, non-maleficence, and justice are currently embedded in the research designs, which obligate researchers to disseminate research findings [7,8,9]. This comes with the debate of which findings to give participants and their clinical significance. The validity and usefulness of the results are the primary concerns for not returning study findings to participants [5,10]. The harm and the risk associated with the dissemination of research findings must be evaluated by researchers in comparison to the benefits. Researchers must assess the harm and risk vs benefits of disseminating research results [11,12,13]. Some of the reported concerns include cost, the burden of subsequent clinical evaluations, potential harm from unnecessary procedures, and emotional stress to participants and family members when findings are uncertain [11,12,13]. Also, physicians might be burdened with explaining findings of unknown significance [11,12,13]. Irrespective of these concerns, physicians and managers are of the view that research findings facilitate evidence-based patient management, enhanced service delivery and pathways, and the transformation of the local culture to enhance patient-centered care [1,14,15].

Studies have reported participants’ interest in receiving research findings [16,17,18]. Dissemination of research results helps participants change their habits and adopt healthier life choices. Satisfied study participants tend to recruit more community members into future studies after they have seen the benefits [16,17,18].

In most cases, researchers do not go back to areas where they conducted studies/collected data to give feedback [19,20]. The concept of engaging the community in the results of research or returning individual results to the respective participants seems more of an afterthought than an integral part of research processes [11]. This leads to a situation in which the research findings benefit researchers rather than the communities from which the data was obtained. In addition, studies conducted on the subject of the dissemination of the results to participants are largely in developed countries [1,14,16,18]. As complex as it is to return results to participants, it is also complex what participants make of those results. There remains a notable gap in studies assessing the impact of research findings on the lived experiences and health of participants, especially in rural South Africa. The current study aims to assess the impact of research results feedback among study participants from a rural black community of low socio-economic status.

## 2. Materials and Methods

### 2.1. Study Design

The present study was qualitative and designed as a narrative research study. The core rationale for this is that narrative analysis is a more full and comprehensive process. It aids scholars not just in developing a deeper comprehension of their subject but also in understanding why people behave in certain ways. The consolidated criteria for reporting qualitative research (COREQ) guiding principle was used in reporting the present study findings [21,22].

### 2.2. Sampling and Study Setting

Data was collected through qualitative face-to-face in-depth interviews using an interview guide. Data was collected during the AWI-Gen 2 research findings feedback session held in the Dikgale area. The Dikgale area forms part of villages that fall under the Dikgale Mamabolo Mothiba Population Health Research Centre (DIMAMO PHRC) surveillance area. The DIMAMO site is located in rural areas about 30 km north of Polokwane, Limpopo province, South Africa (Figure 1) [23]. All participants were of low socio-economic status, and this has been reported elsewhere [24,25]. However, in summary, most participants in the AWI-Gen study where unemployed and had low educational attainment [24,25]. The following criteria were employed to recruit the participants: participants who took part in the AWI-Gen 2 study, and male or female aged 40 years or above. Those who did not participate in AWI-Gen 2, were under the age of 40 years, or did not consent to participate in the study were excluded (Figure 2). The Africa Wits INDEPTH Partnership for Genomic Research (AWI-Gen) is a population cohort that was initiated in 2012. The main goal of AWI-Gen is to investigate how genes, environment, and lifestyle affect cardiovascular and metabolic health across Africa and to uncover new insights that could help to improve the health of Africans [26]. To capture a wide range of perspectives relating to the impact of research results on the lived experiences of study participants, a heterogeneous sampling method was used. Heterogeneous sampling is a purposive sampling technique used to capture a wide range of perspectives relating to the topic that one is interested in studying. This technique seeks to search for variations in perspectives, ranging from conditions that are viewed to be typical to those that are extreme.

### 2.3. Research Questions

The main research question of the study was: What is the perceived impact of results feedback among research participants? The sub-questions included the following:How did participating in research inform your health and lifestyle choices?What were the benefits, if any, of participating in health research and receiving personal findings?What is your view of health research?

**Figure 2 ijerph-22-01565-f002:**
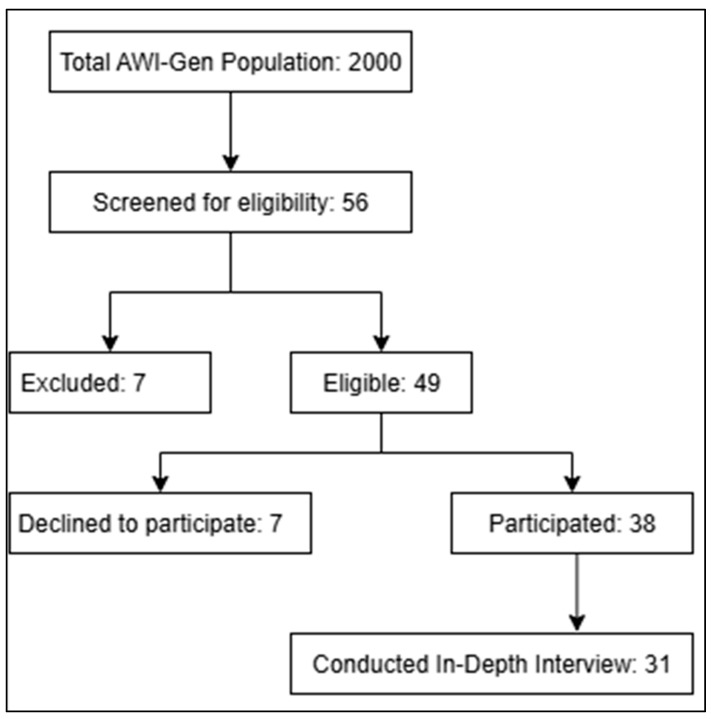
Flow diagram of participant selection.

### 2.4. Data Collection and Analysis

Data was collected through qualitative face-to-face in-depth interviews using an interview guide between September and October 2023 in the Dikgale area of Limpopo province, South Africa, during the AWI-Gen 2 findings feedback and community engagement process. Three principal authors performed semi-structured one-on-one interviews in a private room, with each participant using the primary local language in the region, Sepedi. Three authors conducted the interviews in Sepedi, which is the local language and is also used by the healthcare professionals in the area. The authors clarified the reason for conducting the interviews, the voluntary nature of participation, and the freedom to withdraw at any moment prior to conducting the interview with the participants. Sessions were recorded using audio recorders, with each session lasting around 40 min. Data collection continued until saturation was achieved with participant number thirty-one. Two authors translated and transcribed the interview’s audio recordings word for word. The transcripts were anonymized before being entered into QSR NVivo 10 (QSR International, Warrington, UK) to aid in the analysis. The interview data was analyzed using a deductive content analysis method to develop the themes [27].

## 3. Results

The present study included 31 individuals, of whom 79% were women and 29% were men. All participants were 40 years and older; this is represented in Appendix A (see Appendix A), which shows the sociodemographic characteristics of the included participants. The results section below is organized around themes of the perceptions of participants relating to the impact of research results feedback on the lived experiences of study participants. These themes, identified through deductive analysis, include health and lifestyle awareness, perceived benefits of research participation, community perception of health research, economic and social motivations, and challenges and concerns. These themes and their sub-themes are represented schematically in Figure 3.

**Theme** **1.**
*Health and lifestyle awareness.*


The present study found that giving feedback to participants post participation has a positive influence on the health behavior of the respective participants. This is represented by the sub-themes under theme one, which relates to changes in diet and physical health improvement resulting from participating in research and receiving feedback on their results.

**Sub-theme** **1.1.**
*Dietary changes.*


Some participants reported that, after participating in research and being part of the feedback sessions, which included health talks, they had increased awareness of what to eat/not eat for a healthy lifestyle. These changes included changes in their dietary habits, such as reducing salt, sugar, and fatty foods. For example, the following are what some participants had to say:

*“I don’t eat meat that much, reduced sugar and fat consumption, because I’m the kind of person who eats a lot of vegetables like spinach, cabbage, and potatoes, and I eat meat products at a minimum. I was eating them before, but after participating in research, I had a better understanding that I had to reduce the amount of fat, sugar, and salt that I consume”.* (Participant 1, female, 55–59 years)

*“They perform blood tests on us and test and tell us our health condition, and they advise us on what not to eat and what kind of food to reduce”.* (Participant 14, female, 60–64 years)

When asked if they experienced any health improvements, one participant said the following:

*“Yes, to improve, one has to adhere to treatment from the clinic, eat healthy food like green apples, don’t eat salty foods, don’t consume sugar the food advice needed we are provided at the clinic”.* (Participant 15, female, 65–69 years)

**Sub-theme** **1.2.**
*Physical health improvements.*


Some participants reported that after blood tests, they started feeling physically relieved or healthier, or became more active post-participation. This is due to the held belief that the removal of blood from the system offers some relief from symptoms such as dizziness and headaches.

*“I saw the importance because I had too much blood, but after my blood was withdrawn, my blood was reduced and I felt better”.* (Participant 3, female, 50–54 years)

*“I have participated in research, yes, I can say there is change because when they withdraw blood, it gives me relief in my body and I get better”.* (Participant 4, female, 60–64 years)

*“I see the importance because I had a problem with dizziness, but after the researchers took a bit of my blood for testing, I no longer feel dizzy. Even now, after they took my blood, I felt very relieved, I no longer have a headache also”.* (Participant 26, male, 50–54 years).

**Theme** **2.**
*Perceived benefits of research participation.*


The second theme that emerged from the analysis related to the benefits that the participants perceived from participating in the research study. These relate to access to health information and referrals, which informed how they managed their personal health. The following sub-themes expand further on this.

**Sub-theme** **2.1.**
*Access to health information and referrals.*


Participants indicated that they acquired knowledge about underlying conditions as a result of their participation in research. Additionally, the participants appreciated the referral system that the research center had with primary healthcare facilities, which made it easier to seek further medical assistance post participation in research. This was especially the case when an abnormality was identified.

*“Yes, the disease that I have, I found out about it from participating in research. They told me the results and referred me to the clinic for treatment. Even today I am still following the advice and taking treatment”.* (Participant 7, female, 55–59 years)

*“I felt much better after they took my blood. I felt much better, but I have a problem with my legs. I think it’s because of old age; my knees and joints are very painful. I think the gel in the joints is depleted. What I need is to go to the clinic to get bone medication, and even if I drink a lot of water because of old age, my digestion is not ok because when I go to urinate, my urine is painful, but I noticed it’s old age”.* (Participant 21, female, 70–74 years).

**Sub-theme** **2.2.**
*Personal health management.*


Participants felt better equipped to manage their health conditions, especially with lifestyle changes and dietary advice. Some participants further mentioned that they learned about the diseases they were suffering from as a result of participating in research. Some of the participants indicated that they were able to control their hypertension and blood sugar conditions because of the information that they received by participating in research. This was attributed to a lack of routine checkups and extensive health talks in primary healthcare facilities. Thus, the information they received from the research proceedings and the feedback session was helpful. The following quotes attest to this:

*“I followed the clinic visits so that they could check my blood sugar level, but most of the things I was told by the researchers to tell the truth, I was just having signs that am sick maybe it was because of the little knowledge. I have for medical conditions because when I used to visit the clinic they never told me about diabetes because they never tested my blood sugar… They are very important because they give us important information on how to take care of our lives if you don’t have enough information you can end up being dead”.* (Participant 28, male, 55–59 years).

*“yes after participating I noticed that they hypertension and blood sugar condition are ok and low as I sleep very well because I managed this condition very well”.* (Participant 16, female, 60–64 years).

**Sub-theme** **2.3.**
*Increased health monitoring.*


For some participants, the research provided a form of health monitoring, which they felt was more thorough than regular clinic visits. Some of the tests, which were not originally available on a routine basis in the clinics, were found to give comfort to the participants. Some of the participants also mentioned that the doctors no longer touched them, a factor that made them unsatisfied.

*“Yes, when it comes to participating and you use the machines to scan and explain, even when one was sick, they no longer feel sick because they are just satisfied with the extensive tests that they receive. This is not the case when we go to clinics because when you consult they just ask you a few questions they don’t even put that thing that hangs on the necks on you and old people find comfort in such small gestures, if you put that thing on them and say breath they get satisfied”.* (Participant 12, male, 70–74 years).

*“Yes, I’m very happy with it because they gave me what I did not have, and I’m satisfied because even the testing that took place in campus is more advanced than those in the clinics”.* (Participant 20, female, 55–59 years).

**Theme** **3.**
*Community perception of health research.*


**Sub-theme** **3.1.**
*Value of research in the community.*


There was a generally positive perception towards research activities, and participants perceived research as beneficial for the community’s general health awareness.

*“I feel this benefits the community and I appreciate the help provided by researchers”.* (Participant 21, female, 70–74 years).

*“It cuts us the trip of going far for testing, they come at our homes and do call us to come to the campus for participation”.* (Participant 30, female, 45–49 years)

*“They can come check my blood pressure regularly so that I can know about my life, even the machines used for testing are not there in the clinics”.* (Participant 20, female, 55–59 years)

**Sub-theme** **3.2.**
*Concerns and suggestions for improvement.*


Some participants raised issues such as the desire for more frequent testing, feedback on a wider range of health issues, or culturally sensitive practices by researchers. Some of the diseases that the participants wished could be investigated through research were diseases that involve bones.

*“I think they should check also the bones because I myself have a problem with my back because it is painful”.* (Participant 30, female, 45–49 years)

**Theme** **4.**
*Economic and social motivations.*


**Sub-theme** **4.1.**
*Incentives and compensation.*


The issue of benefit sharing between the researcher and the participants also came out in this analysis. Some participants were appreciative of the financial and logistical support that we provided during the research process. This included the food that was provided for them on days that they were participating in research as well as the transport compensation and/or remuneration that they received. However, they wished that the remuneration could be increased to better support their social and economic needs. Some participants went further to mention that they used the remuneration funds to support their family’s needs. This was especially the case in families that did not receive a government social grant.

*“They must come and test the blood because they don’t take a lot of blood like the blood donation group because we are old and what makes us happy is that they gave us money and food which makes a difference in our lives”.* (Participant 14, female, 60–64 years)

*“… I participated twice (2) in the research in the campus because they give us percent like money, transport and food … Yes it is important because our wives are very string so the money can buy us things we need like… No the incentives is not enough if it is increased it will be fine”.* (Participant 17, female, 55–59 years)

*“… They gave us food like meat and a sum of money and we wish that they could take us back for participation so that the amount can be increased”.* (Participant 18, male, 65–69 years)

*“… Incentive money is important because like myself I am not yet getting old age social grant so the money is important even though it is not enough I wish it can increase because taking care of a household without social grant money is very hard”.* (Participant 19, female, 45–49 years).

*“… I think but it’s up to the researchers, but if I get something that is better it can help me there and there before I did not think of getting paid because my health is more important than money”.* (Participant 20, female, 55–59 years).

*“…it should continue because the money does help us we always have something to do with it and it must be increased because our children are not working”.* (Participant 25, male, 60–64 years).

**Sub-theme** **4.2.**
*Social influence on lifestyle choices.*


Some participants noted how the research encouraged them to influence family and friends towards healthier lifestyles, which reinforced the social value of health research. Further, they mentioned the need to integrate social workers to better serve the community.

*“According to how I view my community they should send social workers to evaluate whether children in the household if given support. People have social problems they don’t even eat even if they get a social grant”.* (Participant 20, female, 55–59 years).

**Theme** **5.**
*Challenges and concerns.*


**Sub-theme** **5.1.**
*Skepticism and misconceptions.*


A few participants expressed concerns about the physical effects of blood sampling or misunderstandings about the cause of certain health conditions. Some participants mentioned feeling weak and developing dry skin as a result of donating blood.

*“Yes, I have seen the importance of research but since my blood was taken I see my skin getting dry and I thought is the blood they took for testing… I have a problem with taking blood because, as I mentioned above that I have skin problem, the skin become dry after donating blood”.* (Participant 15, female, 65–69 years).

*“Yes, the relationship has changed just these days I get tired easily but I feel much better this today my hypertension is better”.* (Participant 23, female, 60–64 years).

**Sub-theme** **5.2.**
*Feedback process and privacy.*


While most were satisfied, a few participants highlighted the need for clear, private communication regarding their health conditions and feedback on results.

*“I don’t have a problem or any concern because when you are giving us results you don’t disclose names and our status…privacy and confidentiality is important for researchers to…”* (Participant 15, female, 65–69 years).

## 4. Discussion

The current study aimed to assess the impact of research findings and feedback among study participants from a rural black community of low socio-economic status. We identified through deductive analysis that engaging the community in research findings had a positive impact on health and lifestyle awareness, perceived benefits of research participation, and community perception of health research. In addition, there were some economic and social motivations to participate in research, which speaks to benefit sharing.

Individuals who participate in academic research from low socioeconomic communities complain of a lack of feedback, unaffordable interventions, and a lack of appreciation for their contributions to research [28]. This results in a perception that research has little to no noticeable impact on the lived experiences of these communities. At the same time, researchers may be more interested in academic expectations such as publication records, thus limiting their time to develop jargon-free, locally tailored (use of local language), cost-effective dissemination materials for nonscientific audiences [28]. This leaves a gap between the knowledge generated and the dissemination, implementation, and impact on the community. The findings of the present study indicate that providing research findings feedback to participants’ post participation has a positive influence on the health behavior of participants.

This was emphasized by the participants’ improvements in physical health and diet changes as a result of participating in research and receiving feedback. The current study observed a variety of lifestyle changes, including modifications to eating patterns, such as a reduction in salt, sugar, and fatty meals. There is solid evidence that engaging the community in evidence-based interventions has a positive impact on a range of health outcomes across various conditions, especially in disadvantaged populations [29]. Although it is not clear how it works, feedback is thought to offer important information, create a sense of a caring and helping relationship to increase engagement in the materials and increase motivation [30]. Nevertheless, the current investigation discovered that a number of participants reported experiencing physical relief, improved health, or an increase in physical activity following the blood tests.

It seems the actual results are somewhat secondary to the perceived relief the participants received from their symptoms after donating blood. This is due to the held belief that the removal of blood from the system offers some relief from symptoms such as dizziness and headaches, as reported in a previously conducted study [31]. However, a few participants expressed concerns about the physical effects of blood sampling, with some participants stating that they felt weak and developed dry skin as a result of donating blood. Although the utmost care is taken in conducting healthcare, in unforeseen cases of harm support and referral are given to the participant.

The second theme highlighted the perceived benefits of research participation, which included access to health information and referrals that helped participants manage their health. They valued the research center’s referral system with primary healthcare facilities, which facilitated further medical assistance. The present study further found that some participants learned through the health research of conditions that had gone undiagnosed. The early diagnosis may have reduced their risk of factors associated with late diagnosis, such as rapid clinical progression of the disease, avoidable morbidity, and mortality [32,33]. The participants highlighted that they do not have access to routine check-ups or individual-focused health talks in primary care. This is because primary healthcare, especially in rural areas, is still plagued by barriers such as poor infrastructure (digital and physical), which forces the health practitioners to triage based on the severity of diseases, with less emphasis on routine testing [34]. However, the participants appreciated the thorough health monitoring provided by the research, which included tests not typically available in regular clinics, offering them comfort and reassurance. The DIMAMO population research center conducted different studies, including the AWI-Gen study, which measured factors such as carotid intima-media thickness, lung function test, and kidney dysfunction, which are not routinely measured in primary healthcare [25].

Although not a diagnostic study, the referral system and relationship with primary healthcare offers an opportunity to refer participants to clinics for further investigation whenever abnormal results are found. This offers early detection of conditions that could be missed in normal healthcare provision, which leans more towards treatment than prevention [35]. In addition, primary healthcare facilities tend to focus on disease treatment rather than routine check-ups due to limited healthcare infrastructure and human resources [36]. Factors such as limited facilities, scattered structures, and insufficient working spaces strain healthcare services, making it challenging to prioritize preventive care [37].

Participants generally had a positive perception of research activities, with some expressing a desire for more frequent and diversified research. However, the issue of benefit sharing/incentives between the researcher and the participants also came out in this analysis. Some participants were appreciative of the financial and logistical support that we provided during the research process. This included the food that was provided for them on days that they were participating in research as well as the transport compensation and/or remuneration that they received. However, they wished that the remuneration could be increased to better support their social and economic needs. Some participants went further to mention that they used the remuneration funds to support their family’s needs. This was especially the case in families that did not receive a government social grant. There has been an increase in recognition of the importance of including benefit sharing in research programs in order to ensure equitable and just distribution of the benefits arising from research [38]. A study by Mwaka et al. [39] reported that benefit sharing should be fair and equitable. This can be done by building capacities and empowering research scientists in developing nations through strengthening regulatory frameworks, extending the purview of the research ethics committee in the development and implementation of material transfer agreements, and meaningfully involving local research communities in benefit-sharing negotiations [39].

### Limitations and Implications of the Study

This research reflects the viewpoints of purposefully selected individuals from the rural black community residing in the DIMAMO area of Limpopo province, South Africa. Consequently, it is focused on a specific population and geographic location and cannot be generalized to other populations. Furthermore, the sample selection of participants was based on those who were present at the AWI-Gen phase 2 results feedback session, excluding those who did not take part in the study. It is noteworthy that the majority of AWI-Gen participants were females (70%), which made it difficult to control for gender differences; hence, the present study has more females compared to males. Nevertheless, the findings of the present study shed light on the impact of research feedback among study participants from a rural black community of low socio-economic status. Given that primary healthcare facilities tend to focus on disease treatment rather than routine check-ups due to limited healthcare infrastructure and human resources, which leads to the prioritization of cure over prevention, research projects could supplement this by providing early detection of some conditions.

## 5. Conclusions

The present study shed light on the importance of retaining individual results for participants in health research. The themes that were generated showed that participants altered their lives and health-seeking behavior because of the information they received from participating in research. Therefore, the concept of engaging the community in the results of research or returning individual results to the respective participants should form an integral part of the research process.

## Figures and Tables

**Figure 1 ijerph-22-01565-f001:**
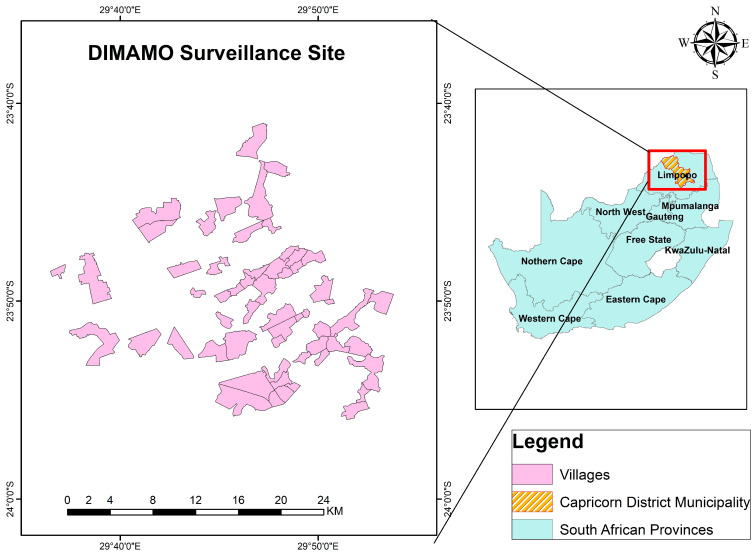
Study area map.

**Figure 3 ijerph-22-01565-f003:**
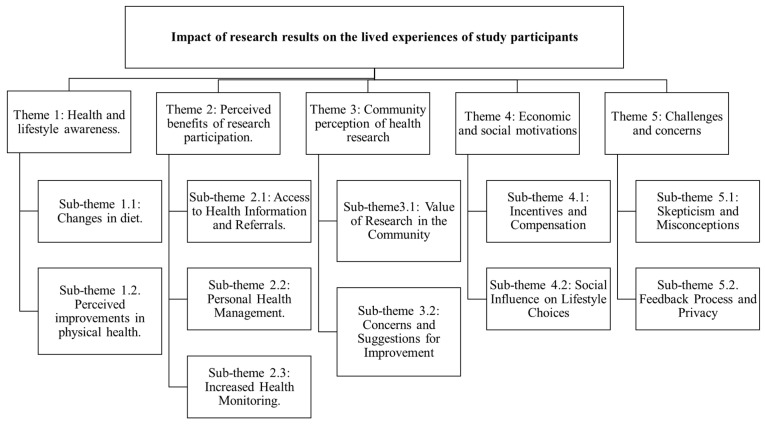
Themes and sub-themes identified through deductive analysis.

## Data Availability

The data presented in this study is available on request from the corresponding author. The data is not publicly available due to privacy or ethical restrictions.

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
