# Peer review of "The Impact of Research Results Feedback on the Lived Experiences of Elderly Participants in the DIMAMO Health Demographic Site: A Case of AWI-Gen Participants"

_ijerph, 2025, doi:10.3390/ijerph22101565_

Round 1

Reviewer 1 Report

Comments and Suggestions for Authors

This is a well-written and valuable manuscript. The topic is highly relevant and addresses an aspect that is often overlooked in current research. The authors’ critical stance, particularly the argument that research should benefit participants and communities, not just researchers, is convincing and important. The empirical materials are rich and well-presented. Overall, this is a meaningful contribution with clear academic merit. I recommend minor revisions before publication.

Specific Suggestions for Revision:

  1. Clarify "AWI-Gen 2 study"
    The manuscript refers several times to the "AWI-Gen 2 study," but never offers a clear explanation of what this study is. A brief definition or contextualisation early in the manuscript would help readers unfamiliar with the project.

  2. Revise the Abstract
    The abstract currently contains excessive methodological detail (e.g., mention of COREQ), while giving too little attention to the actual findings. The balance should be adjusted to highlight the key results and reduce technical references that are more appropriate for the methods section.

  3. Rework Figure 2
    Figure 2 lacks visual clarity and analytical strength. The three branches presented are overlapping rather than mutually exclusive, which makes the figure misleading. A written explanation might be a more effective format to present this structure.

  4. Remove Figure 3
    Figure 3 adds little value to the manuscript. Its content could be conveyed clearly in a single sentence. Unless it can be significantly improved to carry analytical meaning, it is better removed.

  5. Avoid Author-Centric Language in Methods
    The methods section includes some sentences that describe what the individual authors did. These should be removed or rephrased in a neutral, non-author-centred tone. Details about specific author roles can be left to the "Author Contributions" section at the end

Author Response

Dear Reviewer 
Thanks for your inputs 

Please find the attached table of corrections 

Best regards  

Reviewer 2 Report

Comments and Suggestions for Authors

This qualitative study addresses a critical gap in health research practices by examining the impact of providing research results feedback to participants from a rural, low socio-economic rural black community. The researchers adopted a qualitative methodology, conducting in-depth interviews with 31 participants (predominantly women at 79%) and applying deductive content analysis. Findings reveal five key themes demonstrating the transformative potential of results sharing with community. These include: participants reported increased health and lifestyle awareness, recognised tangible benefits from research participation, developed more positive perceptions of health research, experienced various economic and social motivations, while also identifying implementation challenges. 
This research study provides compelling evidence that participants may be likely to modify their health-seeking behaviours and lifestyle choices if they receive individual research results. These changes would not have occurred without this communication cycle. This research makes an important contribution to the literature by demonstrating that results sharing should be viewed as an integral component of the research process rather than an optional afterthought, particularly when working with underserved communities who may derive substantial benefit from understanding their health data.

A few points for the authors' consideration:

2.4 Data collection

Line113-4  How did the researchers determine saturation at participant 31?

I question this as there are only 4 male participants.

2.2 Sampling and setting

The abstract states that the community in this is low socioeconomic, could the authors state in this section to reclarify that all participants were of low socio-economic status.

Line 118 A citation for deductive coding should be included.

  1. Results

Line 122 delete “about” as there were exactly 31 participants.

Line 126  Table 1 contains only gender and age for demographic characteristics. Were any other characteristics considered for this study? Could educational status be included in Table 1? Is there a gender difference in educational status? If so, there are not enough male [participants to make this claim.

  1. Results

Most of the qualitative data shared in quotes from the participants is from female participants.  There should be more quotations from more male participants.

There is no statement about limitations of this study. It seems the very small number of male participants is a limitation in this study. Could the authors clarify this in section 2.4 as requested, and again in a statement of limitations?

Author Response

Dear Reviewer
Thanks for your inputs 

Please find the attached table of corrections 

Best 
